# Bouncing Dynamics of Impact Droplets on the Biomimetic Plane and Convex Superhydrophobic Surfaces with Dual-Level and Three-Level Structures

**DOI:** 10.3390/nano9111524

**Published:** 2019-10-25

**Authors:** Zhongxu Lian, Jinkai Xu, Wanfei Ren, Zuobin Wang, Huadong Yu

**Affiliations:** 1Ministry of Education Key Laboratory for Cross-Scale Micro and Nano Manufacturing, Changchun University of Science and Technology, Changchun 130022, China; lianzhongxu@cust.edu.cn (Z.L.); xujinkai2000@163.com (J.X.); renwaifei@ccut.edu.cn (W.R.); wangzb@cust.edu.cn (Z.W.); 2International Research Centre for Nano Handling and Manufacturing of China, Changchun University of Science and Technology, Changchun 130022, China

**Keywords:** wire electrical discharge machining, aluminum alloy, superhydrophobic property, bouncing dynamics of impact droplets, contact time

## Abstract

Reducing the contact time of a water droplet on non-wetting surfaces has great potential in the areas of self-cleaning and anti-icing, and gradually develops into a hot issue in the field of wettability surfaces. However, the existing literature on dynamic behavior of water drops impacting on superhydrophobic surfaces with various structural shapes is insufficient. Inspired by the microstructure of lotus leaf and rice leaf, dual-level and three-level structures on plane and convex surfaces were successfully fabricated by wire electrical discharge machining on aluminum alloy. After spraying hydrophobic nanoparticles on the surfaces, the plane and convex surfaces with dual-level and three-level structures showed good superhydrophobic property. Bouncing dynamics of impact droplets on the superhydrophobic surfaces wereinvestigated, and the results indicated that the contact time of plane superhydrophobic surface with a three-level structure was minimal, which is 60.4% less than the plane superhydrophobic surface with dual-level structure. The effect of the interval *S*, width *D*, and height *H* of the structure on the plane superhydrophobic surface with three-level structure on contact time was evaluated to obtain the best structural parameters for reducing contact time. This research is believed to guide the direction of the structural design of the droplet impinging on solid surfaces.

## 1. Introduction

Water droplet impact on solid surfaces is commonly found in our daily lives. It is a very important physical phenomenon that reflects the dynamic wetting behavior of water droplets on solid surfaces. With the deepening of research on superhydrophobic surfaces in recent years, researchers have found that reducing the contact time of a water droplet on superhydrophobic surfaces plays an important role in self-cleaning and anti-icing applications and gradually develops into a hot issue in the field of wettability surfaces [1,2,3,4,5,6,7,8,9]. Superhydrophobic surface with reduced contact time can be attributed to a macro-structure on the surface or some new materials [10,11,12,13].

Up to now, various fabrication methods of superhydrophobic aluminum surfaces have been proposed, such as chemical etching [14,15,16,17], electrochemical etching [18,19], anodization [20,21], laser irradiation [22,23,24], electrospinning [25], layer-by-layer assembly [26], and spraying method [27]. Although the above methods can achieve the superhydrophobic aluminum surfaces, the researchers are only interested in the wettability of the surface or the preparation of superhydrophobic surfaces, and little attention is paid to the dynamic contact process of the droplets on the superhydrophobic surface. If we can reduce the contact time on superhydrophobic aluminum surfaces, the range of use of superhydrophobic surfaces will be much expanded.

In general, by means of adding macro-scale structures on a flat superhydrophobic surface, the asymmetric bouncing would appear, which can effectively reduce the contact time of the water droplet on the surface. In 2013, Bird et al. investigated the method for reducing contact time and they found that the contact time of the water droplet could be shortened by asymmetric superhydrophobic surfaces with ridges [28]. Since then, Song et al. and Guo et al. conducted similar experiments to study the influence of various macro-scale structures on the bouncing dynamics of the water droplet [29,30]. In addition, some studies on the water droplet impinging on convex surfaces have been reported [31,32,33]. Liu et al. studied the impact of the water droplet on convex surfaces and found that the water droplet impinging on Echevaria leaves exhibited asymmetric bouncing dynamics with distinct spreading and retraction along with two perpendicular directions [31]. Zhang et al. studied the impact behavior between water droplets with different velocities on cylindrical superhydrophobic surfaces with various diameters, and the experiment results showed that the diameter ratio and Weber number were the determinants of the bouncing patterns [32]. However, the surfaces reported in these studies all have a single structural shape either with a macro-scale structure or just a convex surface, and few of the researches done so far have focused on the combination of the macro-scale structure and convex surface.

In this study, biomimetic plane and convex surfaces with dual-level and three-level structures were successfully fabricated by wire electrical discharge machining (WEDM) on aluminum alloy, and then sprayed with hydrophobic nanoparticles to achieve excellent superhydrophobic property. The morphology, wettability, and contact time of these superhydrophobic surfaces were evaluated. By comparing the bounce behaviors of water droplets on these different topographical surfaces, we obtained the result that the water droplets had the shortest contact time on a plane superhydrophobic surface with dual-level structure. Progressively we chose the plane superhydrophobic surface with dual-level structure as the research object and studied the influence of the interval *S*, width *D*, and height *H* of the structure on the contact time to obtain the best structural parameters. Controlling contact time by changing structural parameters will have a wide variety of potential applications in the areas of heat transfer, anti-icing, and self-cleaning.

## 2. Experimental Section 

### 2.1. Materials and Methods

7075 aluminum alloy was purchased from Shanghai Jiuqin Metal Materials Co, Ltd. (Shanghai, China). Acetone was purchased from Tianjin Xintong Fine Chemical Co, Ltd. (Tianjin, China). Absolute ethanol was purchased from Beijing Chemical Factory (Beijing, China). Never Wet was purchased from Hangzhou Nanyan Trading Co, Ltd. (Hangzhou, China).

The surface microstructures are constructed using a WEDM. Recently, the technology has been widely used to fabricate special wettability surfaces [34,35,36,37,38]. It can achieve the construction of macro-structure, even the processing of convex surface or combination of macro-structure and convex surface. During the WEDM process, the wire working electrode and tool electrode are discharged, and the discharge channel instantaneously generates high-temperature heat energy, so that the workpiece material is etched away, and a micro-crater-like structure is formed on the surface of the workpiece material. The procedure of the WEDM is as follows: The sample was clamped onto the WEDM, and the position of the sample and molybdenum wire (diameter: *Φ*0.18 mm) was adjusted so that the molybdenum wire could be located at the beginning of processing. The programmed program was input into the disk of WEDM, and it needed to adjust the pulse width (24 μs), pulse interval (96 μs), and wire speed (11.6 m/s) and call the input programmed program to start processing. After processing, the sample was taken from the worktable, and the surface of the sample was cleaned with acetone, absolute ethanol, and deionized water to remove the impurities on the surface of the sample, and then dried for reserve. Finally, the as-fabricated sample was sprayed with Never Wet consisting of hydrophobic nanoparticles dispersed in acetone to obtain superhydrophobicity. Illustration of the entire fabrication process is shown in Figure 1. Prior to testing and characterization of as-obtained samples, alcohol was used to rinse those nanoparticles that were not bound to the surfaces.

Scanning electron microscopy (SEM, Quanta 250, FEI, Hillsboro, OR, USA) was used to obtain the microscopic images of the surfaces. Energy-dispersive diffraction (EDS, INCAEnergy, Oxford Ins, Oxford, UK) and Fourier transform infrared (FTIR) spectrometer (Nicolet iS 50, Thermo Fisher Scientific, Waltham, MA, USA) were employed to obtain the chemical composition of the surfaces. The self-made contact angle measuring instrument (JCJ-001, CUST, Changchun, China) was used to obtain the photograph of the droplets, and the contact angle values were obtained by analyzing the photograph with the software (Contact Angle Version 1.0, CUST, Changchun, China). The super-high speed camera (Pco.dimax HS4, PCO, Kelheim, Germany) was used to obtain the bouncing process of water droplets.

Usually, researchers use Weber number *We* (dimensionless dynamic parameters) to study the impact dynamics of water droplets on solid surfaces [29]. The Weber number is the ratio between the inertial force of the water droplet and the surface tension effect, and the Weber number can be expressed as:(1)$$We=\frac{\rho RV^{2}}{\gamma_{lv}},$$
where *R* is the radius of the water droplet, *V* is the velocity at which the water droplet impacts the surface, *ρ* and *γ_lv_* are the density and surface tension of water (*ρ* = 1 g·cm^−3^, *γ_lv_* = 72.1 mN·m^−1^), respectively.

### 2.2. Design of Dual-Level Structure Imitating Lotus Leaf Surface and Three-Level Structure Imitating Rice Leaf Surface

Koch et al. [39] obtained the microscopic morphology of the lotus leaf surface by SEM and found that the surface had a micron-sized mastoid structure and a hydrophobic waxy layer. Feng et al. [40] revealed that the key to obtaining superhydrophobic and self-cleaning properties lies in the construction of micro-nano-scale structures. Their further observation of the surface of the lotus leaf revealed that a branched nano-scale structure with a diameter of about 120 nm was randomly distributed on the micron-sized mastoid structure (Figure 2a,b), which constitutes the micro-nano-scale structure morphology of the lotus leaf surface. Based on the principle of dual-level structure on lotus leaf surface and combined with the characteristics of the WEDM technology, this paper designed the plane and convex surfaces with a dual-level structure on the aluminum alloy substrate (Figure 2c,d).

Another plant that attracts scientists’ attention is the rice leaf. Water droplets flow preferentially along the direction parallel to the edge of the leaf [41,42], and sliding angles parallel to the edge of the leaf and in the vertical direction are about 3°–5° and 9°–15°, respectively. From the SEM images of the surface of rice leaf [43], the existence of sub-millimeter-scale grooves on the surface was observed, exhibiting anisotropic wetting, and the rice leaf surface had a micro-nano-scale structure similar to the lotus leaf surface (Figure 3a,b). It is this three-level complex structure that results in the superhydrophobicity of anisotropic wetting on rice leaf surface. Aiming at the structure characteristics of rice leaf, this paper also designed the plane and convex surfaces with three-level structure on the aluminum alloy substrate (Figure 3c).

## 3. Results and Discussion

### 3.1. Relationship between Surface Microstructure and Wettability

According to the structural characteristics of the lotus leaf and rice leaf surfaces, the dual-level and three-level structures were constructed on the plane and convex surfaces of the aluminum alloy by WEDM technique. Figure 4 shows the SEM images of the surface after mechanical polishing and WEDM processes. It can be seen that the polished surface was very smooth and only had shallow scratches (Figure 4a–c). Figure 4d shows the microscopic morphology of the WEDM plane surface with dual-level structure at low magnification. Micron-scale crater structures with diameters ranging from 30 to 60 μm could be seen. In addition, many mastoid structures appeared randomly around the crater structures. By further magnifying the SEM image (Figure 4e), there were many inconspicuous nano-scale structures on the crater and mastoid structures. Figure 4g–i is the morphology of the WEDM plane surface with three-level structure. By fabricating the sub-millimeter-scale groove structure designed above and combining with micro-nano-scale structure formed spontaneously during processing, the three-level structures (sub-millimeter, micron and nano-scale structures) were constructed. Figure 4g is a SEM image of the WEDM plane surface with three-level structure at low magnification. It is found that the sub-millimeter-scale groove structure on the WEDM surface was uniformly aligned (the width and height of the groove structure were approximately 82 ± 3.8 μm and 365 ± 9.7 μm), and the groove structure was covered with micro-scale crater and mastoid structures. Further enlargement (Figure 4h) shows that there were many inconspicuous nano-scale structures on the crater and mastoid structures on the surface of the aluminum alloy, which is similar to the above-described dual-level structure. After cutting the convex surface of aluminum alloy according to the prescribed procedure, it is seen that dual-level structures with obvious micro-crater and mastoid structures (Figure 4j) and inconspicuous nano-scale structure (Figure 4k) were obtained on the surface of the sample, and the dual-level structure on the WEDM convex surface was basically the same as that on the plane surface. Figure 4m,n is SEM images of the WEDM convex surface with three-level structure. It is observed that the convex surface was distributed with a sub-millimeter-scale groove structure (the width and height of the groove structure were approximately 85 ± 6.2 μm and 398 ± 3.7 μm, as shown in Figure 4m), the groove structure was also covered with micro-scale crater and mastoid structures and inconspicuous nano-scale structures (Figure 4k). Figure 4c,f,i,l,o is cross-sectional SEM images of polished surface, plane surface with dual-level structure, plane surface with three-level structure, convex surface with dual-level structure, convex surface with three-level structure, respectively.

It can be seen from the above that the plane and convex surfaces with obvious micro-scale crater and mastoid structures and inconspicuous nano-scale structures were fabricated on aluminum alloy, which are in good agreement with dual-level structure designed in this paper. By cutting the sub-millimeter-scale groove structures on the surface of the aluminum alloy, combined with the micro-scale crater and mastoid structures and the inconspicuous nano-scale structures, the three-level structures of the plane and convex surfaces was constructed, which are in good agreement with the designed three-level structures.

It has been reported that water droplets can only achieve hydrophobicity in the plane surface with a dual-level structure [44], which makes it impossible to systematically study and compare the effects of different surface morphologies on the dynamic behavior of water droplet impact. Therefore, in order to achieve the superhydrophobicity of the above-mentioned WEDM plane and convex surfaces with dual- and three-level structures, the Never Wet is used to reduce the surface energy of the sample. The SEM image of the Never Wet coating is shown in Figure 5a. The coating is made up of a large number of nano-scale particles with diameters ranging from 30 to 60 nm. Figure 5b,c is the EDS and FTIR results of the coating. The elements silicon, carbon, and oxygen were observed on the surface, and the corresponding weight proportions were 44.9%, 44%, and 11.1%, respectively. The result indicates that the coating is mainly composed of silicon-based nanoparticles.

As shown in Figure 6a, the water contact angle of polished aluminum alloy surface was 69.3 ± 2.8°. Figure 6b–e is the photographs of water droplets on different aluminum alloy surfaces after WEDM and Never Wet treatments. The water contact angles of the plane surfaces with the dual-level and three-level structures were 152.7 ± 1.5° and 156.9 ± 0.7°, respectively (Figure 6b,c). For the wettability measurement of the convex surface, since the water droplet could not stably stay on the surface, the advancing contact angle was used to approximate the contact angle. As shown in Figure 6d,e, the contact angles of water droplet on the convex surfaces with the dual- and three-level structures were 153.9 ± 2.4° and 155.4 ± 1.6°, respectively. It should be noted that by comparing the hydrophobicity of the WEDM surface, the contact angle of water droplets in the plane surface with dual-level structure is larger than that with three-level structure, and for the convex surface, the contact angle of dual-level structured surface is also larger than the three-level structured surface.

Under the condition of ignoring the influence of the gravity on the shape of water droplet, the contact angle is related to the diameter of the solid–liquid contact surface (the length of the three-phase contact line), and the contact angle *θ* of the solid surface can be expressed as [45]:(2)$$\theta =2\arctan\frac{2h}{l},$$
where *h* is the height of the water droplet and *l* is the length of the three-phase contact line. For a water droplet of 4.3 μL, the length of the three-phase contact line measured was about 1.07 mm when it was placed on the plane surface with dual-level structure, and the length of the three-phase contact line of the water droplet on the plane surface with three-level structure was about 0.9 mm. Therefore, in the case where the heights of the two droplets are almost the same, the contact angle of the dual-level structured plane surface with the long three-phase contact line is small. Similarly, it can be known that the length of the three-phase contact line of the convex surface with dual-level structure is larger than that with three-level structure, so the contact angle of the convex surface with dual-level structure is small.

### 3.2. Analysis of Impact Behavior of Water Droplet 

Figure 7 shows the bouncing process of the water droplet on the polished aluminum alloy surface (drop volume *V* was 4.3 μL, drop height *h* was 10 mm, Weber number *We* was 2.8). It is observed that after the water droplet was in contact with the surface, although there was a tendency to move upwards, it was always unable to bounce and finally stabilized on the smooth aluminum alloy surface, because the smooth aluminum alloy surface had hydrophilicity (Figure 6a).

So far, the pancake-like bouncing of water droplets is the most effective way to achieve rapid bounce. Water droplets bounce off the surface directly when spread to the maximum radius, and there is no need to undergo a retraction process before the droplets bounce off [34]. Generally, a pancake-like bouncing is defined as the ratio (*Q*) of the transverse extension diameter (*d_jump_*) of a water droplet leaving the surface to the maximum transverse extension diameter (*d_max_*) of the water droplet during the jump greater than 0.8, that is *Q* = (*d_jump_*/*d_max_*) > 0.8, as shown in Appendix A. Unlike traditional superhydrophobic surfaces with small roughness, when a water droplet is in contact with a surface having a large-scale columnar microstructure, a portion of the water droplet immerses between the arrays for energy storage. When the water droplet rebounds, the capillary energy penetrating into the structure is rectified into enough kinetic energy to lift the whole droplet upward, and the droplet is pushed directly off the surface.

This paper studies and compares the bouncing behavior of water droplets on dual-level structured plane surface, three-level structured plane surface, dual-level structured convex surface, and three-level structured convex surface of aluminum alloy obtained by WEDM and Never Wet treatments. Figure 8 shows the bouncing process of water droplets on different surfaces. In the experiment, the following parameters were used unless otherwise specified: The droplet volume *V* was 27.4 μL, the radius *r* was 1.87 mm, and the drop height *h* was 30 mm. According to the formula (1), the Weber number *We* was 15.2. As shown in Figure 8a, the water droplet contacted the dual-level structured plane surface from 0 ms. At 3.9 ms, the center of gravity of the water droplet dropped to the lowest point. Then the droplet began to bounce until 15.9 ms was completely separated from the surface, that is, the contact time *t_c_* was 15.9 ms. By calculation, it is known that *Q* was about 0.45, showing normal bouncing behavior. Figure 8b is the bouncing process of water droplet on a three-level structured plane surface. It can be observed that the time required for the water droplet to bounce off the surface was 6.3 ms, which is 60.4% less than that of the water droplet on the dual-level structured plane surface, and the *Q* is 0.91, which shows typical pancake-like bouncing behavior. As shown in Figure 8c, for a convex surface with a dual-level structure, the time required for the water droplet from contact to complete detachment on its surface was 11.7 ms, which is less than the time on the dual-level structured plane surface, but is higher than the time on the three-level structured plane surface. Figure 8d is a bouncing process of the water droplet on a convex surface with three-level structures. It is seen that the time taken for the water droplets to contact from the surface to the complete detachment was 10.2 ms. Compared with the contact time of the dual-level structured plane and convex surfaces, this time is a reduction, but still more than the three-level structured plane surface. It indicates that compared with the four different surfaces, the contact time *t_c_* of the water droplet on the dual-level structured plane surface is the longest, while the contact time *t_c_* on the three-level structured plane surface is the shortest. Compared with previous studies [46,47], the contact time *t_c_* (6.3 ms) is significantly reduced, and a typical pancake-like bouncing behavior was observed.

Based on the above experimental results, this paper mainly studies the influence of the microstructure size of three-level structured plane surface on the contact time *t_c_* and *Q*. The structure parameters of three-level structured plane surface are shown in Appendix A, where *S*, *D*, and *H* represent the interval, width, and height of groove structure, respectively. In order to retain sufficient capillary energy during bouncing of the water droplet, the height *H* of the groove structure was kept constant at 0.66 mm. Figure 9 shows the variation of contact time *t_c_* and *Q* with the interval *S* of groove structure (Weber number *We* was 15.2, width *D*, and height *H* of groove structure were 0.2 mm and 0.66 mm, respectively). It can be seen that the contact time *t_c_* increased as the interval *S* increased, and *Q* decreased as the interval *S* between the grooves increased. When the interval *S* was 0.28 mm and 0.41 mm, *Q* > 0.8, and when interval *S* was 0.52 mm, 0.62 mm, and 0.72 mm, *Q* < 0.8. Therefore, when the water droplet forms a pancake-like bouncing, the maximum interval *S* of the groove structure was about 0.52 mm, the *Q* was 0.85, and the contact time *t_c_* was 7.8 ms. Figure 10 shows the effect of the width *D* of groove structure on the contact times *t_c_* and *Q* (the interval *S* and height *H* of the groove structure were 0.28 mm and 0.66 mm, respectively) when the Weber number *We* was 15.2. The results show that when the width *D* was within 0.47 mm, the contact time *t_c_* was less than 7.2 ms, and *Q* > 0.8, showing typical pancake-like bouncing behavior. However, for a width *D* of 0.7 mm and 0.88 mm, the *Q* was less than 0.8, the water droplet exhibited a conventional bouncing behavior. Therefore, for a water droplet having a volume of 27.4 μL, the maximum width *D* of the groove structure to form a pancake-like bouncing was 0.47 mm.

It has been reported that for surfaces with a small width, Liu et al. [34] pointed out that the surface of the height *H* under moderate conditions had the phenomenon of pancake-like bouncing. In this paper, a similar law was found when studying the effect of the height *H* on contact time *t_c_* and *Q*. Figure 11 shows the contact time *t_c_* and *Q* of the water droplet (volume *V*, radius *r*, and Weber number *We* were 27.4 μL, 1.87 mm, and 15.2, respectively) on the three-level structured plane surface with different heights *H*, and the interval *S* and width *D* were constant at 0.28 mm and 0.47 mm, respectively. It is observed that there was a pancake-like bouncing behavior on the surface having a height *H* of 0.66 mm to 0.87 mm, and for a height *H* > 0.87 mm or *H* < 0.66 mm, water droplet exhibited a conventional bouncing behavior on the surface.

We also studied the effect of Weber number *We* on contact times *t_c_* and *Q*, and tried to find the minimum *We* that formed the pancake-like bouncing behavior. Figure 12 shows the contact time *t_c_* and *Q* of the water droplet (the volume *V* and radius *r* were 27.4 μL and 1.87 mm, respectively) on the three-level structured plane surface at different intervals with the change of Weber number *We*, and the width *D* and height *H* were constant at 0.2 mm and 0.66 mm, respectively. Here, the Weber number *We* was controlled by changing the drop height *H* of the water droplet. For the interval *S* of 0.72 mm, there was no pancake-like bouncing behavior in any of *We*, and the contact time *t_c_* was greater than 14.4 ms. For the interval *S* of 0.62 and 0.52 mm, pancake-like bouncing occurred at *We* ≥ 16.2 and *We* ≥ 17.2. When the interval *S* was less than 0.41 mm, the critical value of *We* for the water droplet to achieve pancake-like bouncing was 11.1, that is, the pancake-like bouncing behavior occurred when *We* ≥ 11.1. The results show that when the interval *S* was less than 0.41 mm, the water droplet could obtain a pancake-like bouncing behavior under most *We* conditions.

## 4. Conclusions

In this research, inspired by the microstructure of lotus leaf and rice leaf, plane and convex surfaces with dual-level and three-level structures on aluminum alloy were successfully fabricated by WEDM. The wettability and contact time on the surfaces were studied. The contact angles or advancing contact angles of the four surfaces were higher than 150°, showing good superhydrophobic property. Bouncing dynamics of impact droplets on the plane and convex surfaces with dual-level and three-level structures was investigated. The results indicated that the contact time of plane superhydrophobic surface with three-level structure was minimal, which was 60.4% less than the surface with dual-level structure, and the typical pancake-like bouncing behavior was observed on the plane with three-level structure. The effect of structural parameters of plane superhydrophobic surfaces with three-level structure on contact time was studied. In addition, the relationship of Weber number and contact time was established to find the minimum Weber number that formed the pancake-like bouncing behavior. The results show that when the interval *S* was less than 0.41 mm, the water droplet could obtain a pancake-like bouncing behavior at *We* ≥ 11.1. Changing these parameters can control the contact time, which would have a potential application in anti-icing and self-cleaning. This research is believed to enrich the structural shapes of superhydrophobic surfaces with shortened contact time and guide the direction of the structural design of the droplet impinging on solid surfaces.

## Figures and Tables

**Figure 1 nanomaterials-09-01524-f001:**
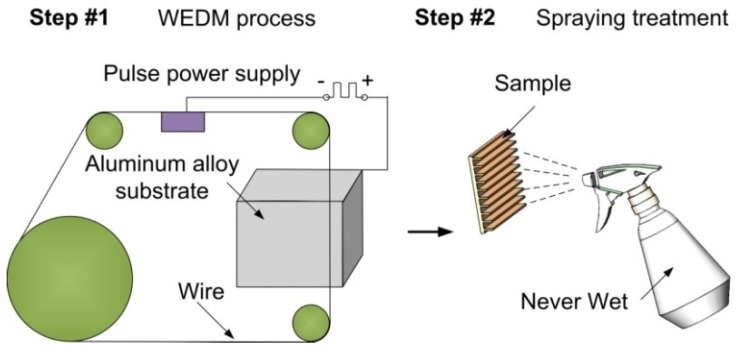
Illustration of the entire fabrication process.

**Figure 2 nanomaterials-09-01524-f002:**
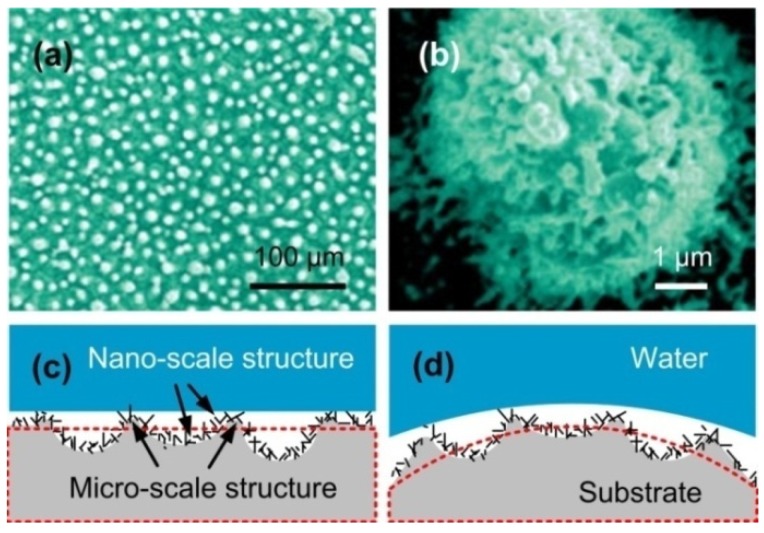
Surface morphology of the lotus leaf and schematic diagram of dual-level structure: (**a**,**b**) Micron-sized mastoid and nano-sized branching structures (Reproduced with permission from [40]. Copyright Wiley-VCH, 2002); (**c**) designed plane surface with dual-level structure; (**d**) designed convex surface with dual-level structure.

**Figure 3 nanomaterials-09-01524-f003:**
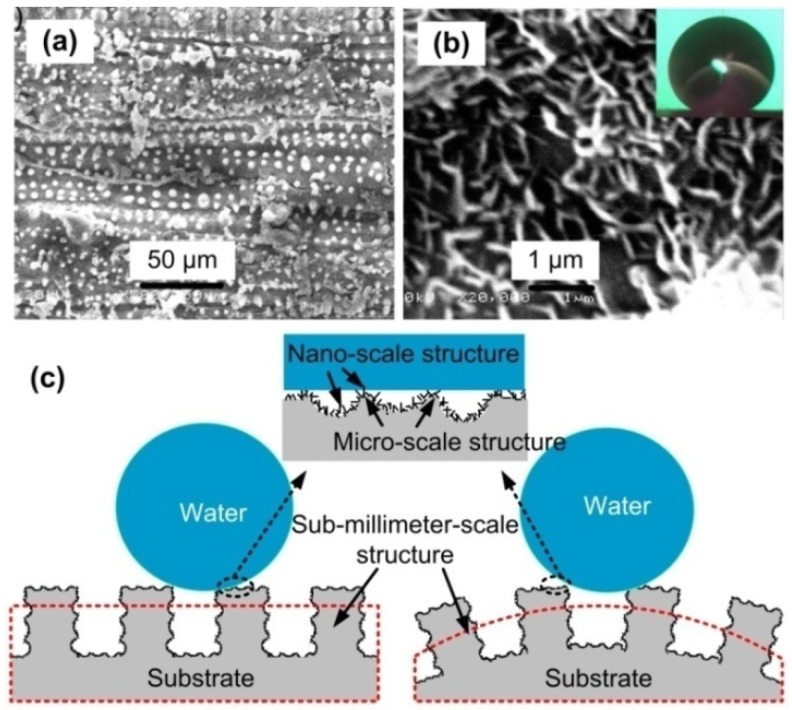
Morphology of rice leaf and schematic diagram of three-level structure: (**a**,**b**) SEM images of rice leaf surface, and the inset of (**b**) is a water droplet on the surface with a contact angle of 157° (Reproduced with permission from [43]. Copyright Elsevier, 2011); (**c**) designed plane surface with three-level structure and convex surface with three-level structure.

**Figure 4 nanomaterials-09-01524-f004:**
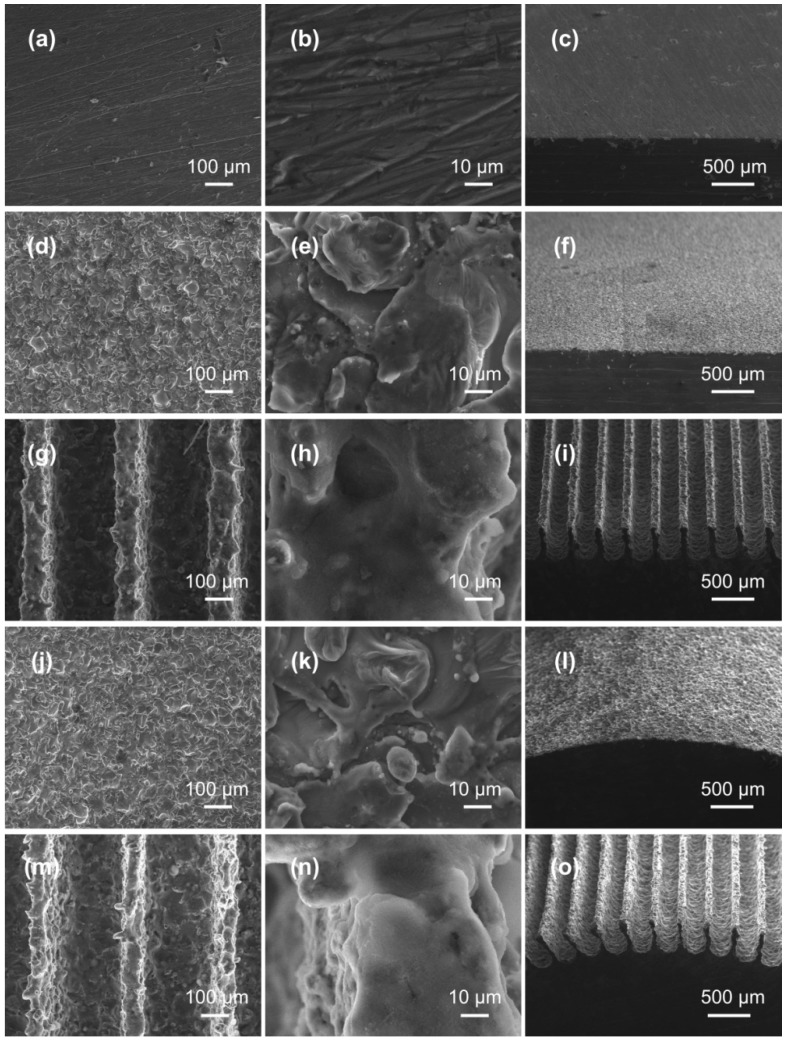
SEM images of different WEDM aluminum alloy surfaces: (**a**–**c**) Morphology and cross section of polished surface; (**d**–**f**) the morphology and cross section of plane surface with dual-level structure; (**g**–**i**) the morphology and cross section of plane surface with three-level structure; (**j**–**l**) the morphology and cross section of convex surface with dual-level structure; (**m**–**o**) the morphology and cross section of convex surface with three-level structure.

**Figure 5 nanomaterials-09-01524-f005:**
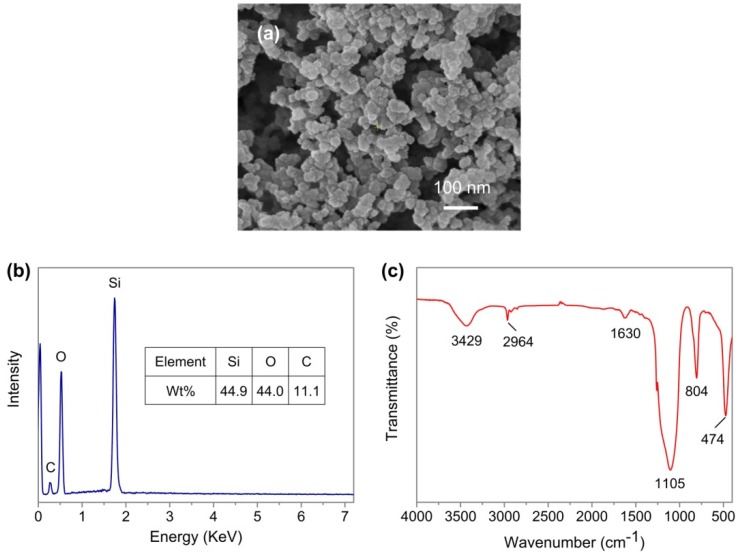
(**a**) SEM image of the Never Wet coating. (**b**,**c**) Energy-dispersive diffraction (EDS) and FTIR results of the Never Wet coating.

**Figure 6 nanomaterials-09-01524-f006:**
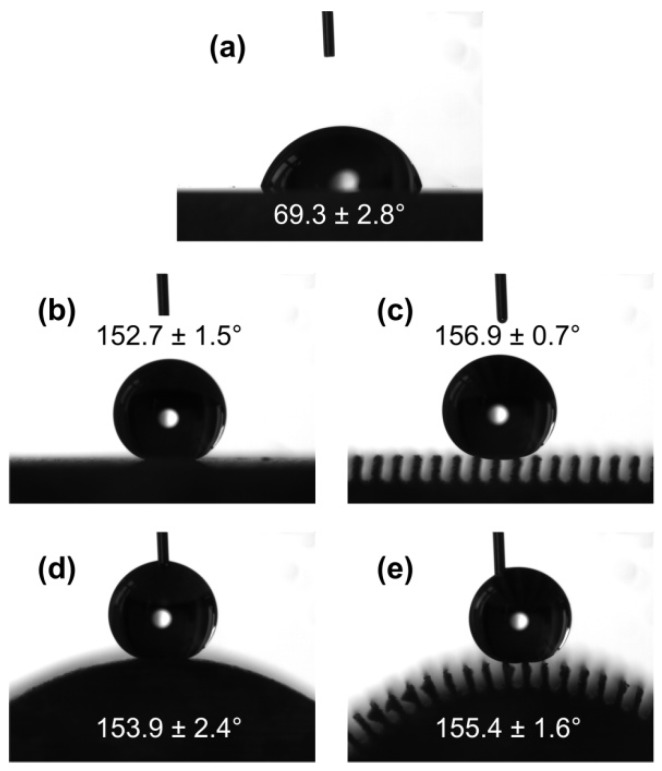
Contact angles of a water droplet on different aluminum alloy surfaces: (**a**) Polish surface; (**b**–**e**) dual-level structured plane surface, three-level structured plane surface, dual-level structured convex surface, and three-level structured convex surface after Never Wet treatment.

**Figure 7 nanomaterials-09-01524-f007:**
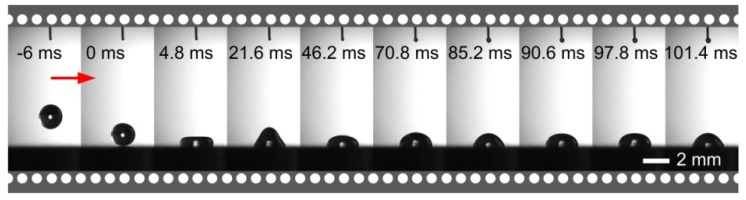
The bouncing process of water droplets on the polished aluminum alloy surface.

**Figure 8 nanomaterials-09-01524-f008:**
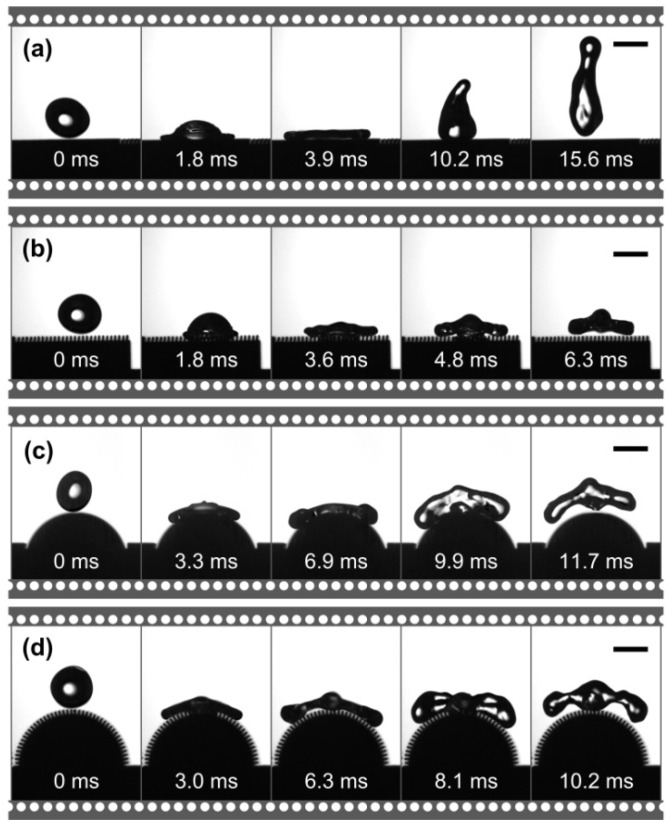
The bouncing process of a water droplet on different aluminum alloy surfaces: (**a**) A dual-level structured plane surface; (**b**) a three-level structured plane surface; (**c**) a dual-level structured convex surface; (**d**) a three-level structured convex surface. All the bars and Weber numbers are 3 mm and 15.2, respectively.

**Figure 9 nanomaterials-09-01524-f009:**
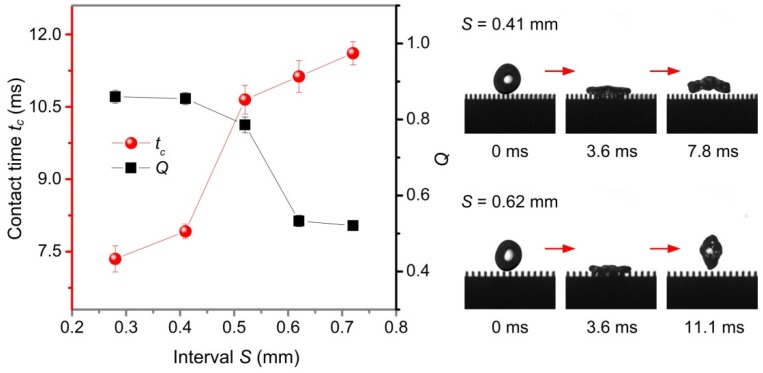
Contact time *t_c_* and *Q* of water droplets in a three-level structured plane surface as a function of interval *S*.

**Figure 10 nanomaterials-09-01524-f010:**
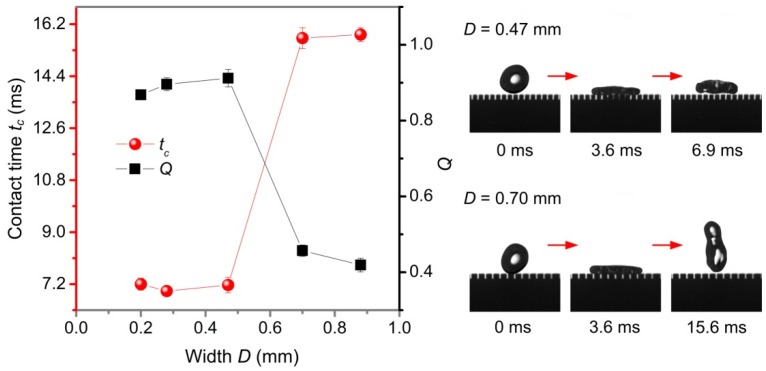
Contact time *t_c_* and *Q* of water droplets in a three-level structured plane surface as a function of width *D*.

**Figure 11 nanomaterials-09-01524-f011:**
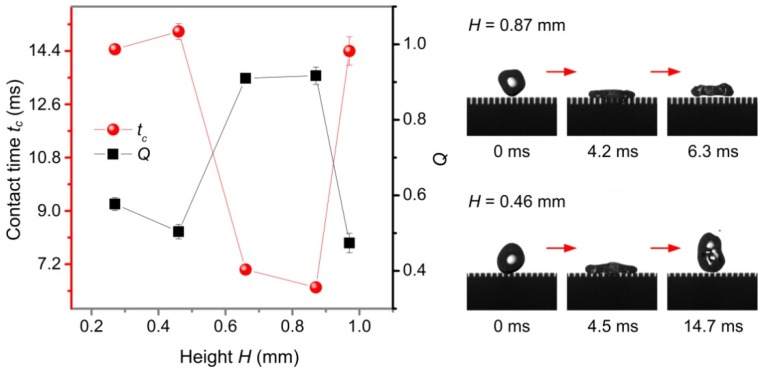
Contact time *t_c_* and *Q* of water droplets in a three-level structured plane surface as a function of height *H*.

**Figure 12 nanomaterials-09-01524-f012:**
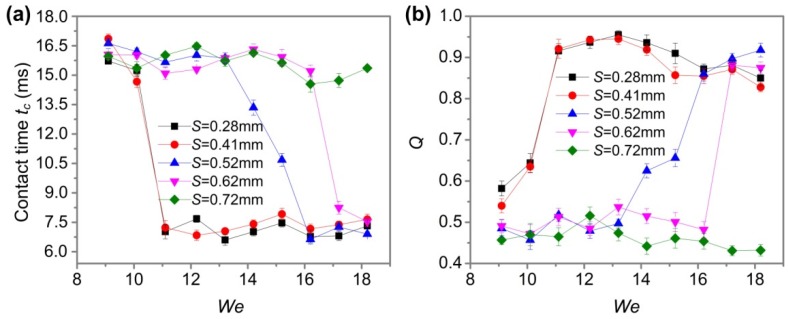
Contact time *t_c_* and *Q* of water droplets at different intervals *S* with the change of Weber number *We*: (**a**) Contact time *t_c_*; (**b**) *Q*.

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
