# Peer review of "Bouncing Dynamics of Impact Droplets on the Biomimetic Plane and Convex Superhydrophobic Surfaces with Dual-Level and Three-Level Structures"

_nanomaterials, 2019, doi:10.3390/nano9111524_

Round 1

Reviewer 1 Report

The manuscript cannot be accepted as it is and needs revision. Below are comments that should help the authors improve the manuscript:

1) Reference 31: the formula, Ti3SiC2, must be presented properly (with subscripts).

2) Line 61: achieved should be replaced with “achieve”? Please double check

3) Figure 5: what nanoparticles were used? Silica? Alumina? Titania nanoparticles? Something else? Please specify.

4) Maybe in caption to Fig.8, the authors should add that all the images correspond to experiments with conditions that provided the same Weber number?

5) There are quite many papers on bouncing droplets (including on superhydrophobic surfaces). Therefore, the authors are recommended to specify in a more clear way, in Introduction, what novel and original this manuscript presents to potential readers. As compared with previously published (numerous) reports.

6) Pages 1-2, Introduction. The authors claim such surfaces can be used for anti-ice applications but do not give any proper references. The following reports can be used as references (some of them are review papers on the subject):

Progress in Chemistry 2017, 29, 102-118

Progress in Materials Science 2019, 103, 509-557

Langmuir 2011, 27, 25-29

Applied Surface Science 2009, 255, 8153-8157

AIP Advances 2019, 9, 055304

Author Response

The manuscript cannot be accepted as it is and needs revision. Below are comments that should help the authors improve the manuscript:

1) Reference 31: the formula, Ti3SiC2, must be presented properly (with subscripts).

We have modified in the revised manuscript.

2) Line 61: achieved should be replaced with “achieve”? Please double check

The word "achieved" have been replaced with “achieve”.

3) Figure 5: what nanoparticles were used? Silica? Alumina? Titania nanoparticles? Something else? Please specify.

We have added the corresponding characterization in Figure 5b and c, and the result indicates that the coating is composed of silicon nanoparticles.

4) Maybe in caption to Fig.8, the authors should add that all the images correspond to experiments with conditions that provided the same Weber number?

We have added conditions of the Weber number in the caption of Fig.8.

5) There are quite many papers on bouncing droplets (including on superhydrophobic surfaces). Therefore, the authors are recommended to specify in a more clear way, in Introduction, what novel and original this manuscript presents to potential readers. As compared with previously published (numerous) reports.

We have modified in the Introduction.

6) Pages 1-2, Introduction. The authors claim such surfaces can be used for anti-ice applications but do not give any proper references. The following reports can be used as references (some of them are review papers on the subject):

Progress in Chemistry 2017, 29, 102-118

Progress in Materials Science 2019, 103, 509-557

Langmuir 2011, 27, 25-29

Applied Surface Science 2009, 255, 8153-8157

AIP Advances 2019, 9, 055304

These reports have been added to the Introduction.

Reviewer 2 Report

The submitted manuscript deals with the wettability of aluminum surfaces, plane and convex in shape, having dual-level and three-level structures. The surface preparation has been performed by WEDM process; nanoparticles have been sprayed to enhance the superhydrophobicity. The water contact angles have been measured and the bouncing dynamics of water drops have been shown.

In my opinion, the manuscript requires revisions before the publication. The following are my comments:

The aim of the study is not clear and it needs to be clarified. The presented phenomena have been already discussed in other studies. Authors should try to better explain the advantages of both the applied procedures and obtained results. The results discussed in Section 3.1 refer to already published data. For example, Figures 2a and 2b are reported in Ref. 33; Figures 3a and 3b are reported in Ref. 36. This section should be moved in the Introduction. Which kind of nanoparticles has been applied? No detail about their nature or dimensions is reported. Lines 106-108: check the citations. Barthlott is not the first author of ref. 32; Jiang is not the first author of ref. 33. Check thoroughly the text: there are typos and mistakes.

Author Response

The submitted manuscript deals with the wettability of aluminum surfaces, plane and convex in shape, having dual-level and three-level structures. The surface preparation has been performed by WEDM process; nanoparticles have been sprayed to enhance the superhydrophobicity. The water contact angles have been measured and the bouncing dynamics of water drops have been shown.

In my opinion, the manuscript requires revisions before the publication. The following are my comments:

The aim of the study is not clear and it needs to be clarified. The presented phenomena have been already discussed in other studies. Authors should try to better explain the advantages of both the applied procedures and obtained results. The results discussed in Section 3.1 refer to already published data. For example, Figures 2a and 2b are reported in Ref. 33; Figures 3a and 3b are reported in Ref. 36. This section should be moved in the Introduction. Which kind of nanoparticles has been applied? No detail about their nature or dimensions is reported. Lines 106-108: check the citations. Barthlott is not the first author of ref. 32; Jiang is not the first author of ref. 33. Check thoroughly the text: there are typos and mistakes. 

We have modified the relevant content on the aim of the study and explained the advantages of both the applied procedures and obtained results in the revised manuscript. We did not move Section 3.1 to the Introduction, but Experimental Section. If you think this is not appropriate, we will modify it further. The coating is composed of silicon nanoparticles, and we have characterized their nature. We also have modified the citations of ref. 32 and ref. 33.

Reviewer 3 Report

Rough surfaces of controlled topographies resembling the lotus and rice leaves were fabricated by wire electrical discharge machining (WEDM) on aluminum alloy. Surfaces were then treated with a commercial water repellent product and bouncing dynamics of impact droplets on the fabricated superhydrophobic surfaces was investigated.

This is a very good and meticulous study which can be useful for the readers of the Nanomaterials. I have no major comments on the Technical part which is very good. However, the structure of the manuscript should be improved.

My main concern is that the authors do not compare their results with previously published works. In the Results and Discussion section very few previously published works are cited. However, there are several articles which investigate the impact of bouncing drops on superhydrophobic, lotus-like surfaces e.g. Applied Surface Science 257(21):8857-8863; Science China Physics, Mechanics & Astronomy 2014, Volume 57, Issue 7, pp 1376–1381. These two works are provided as examples. There is a considerable amount of literature on this topic, which is neglected by the authors and it is not included in the Introduction. The authors should include these previously published works in the Introduction describing the novelty of their paper. As the goal of this work is to study the bouncing drops, the novelty cannot be the use of wire electrical discharge machining. More important, the authors should compare (Results and Discussion section) their results with the data of the literature.

Likewise, in the Introduction in the paragraph in which the various techniques are summarised it is better to focus on articles describing the achievement of superhydrophobicity on aluminum surfaces (e.g. Langmuir 2008, vol 24, p 11225 and others) trying to avoid papers which are not related to the substrate of interest. Other works which use nanoparticles for superhydrophobicity can be included.

The Abstract and particularly the Conclusion sections are not very informative. In the Conclusion section the conclusions should be described more precisely, avoiding general comments such as for instance, “Different surfaces had different effects on the contact time of water droplet.”

What is the contact angle of a water drop on Never Wet on polished aluminum? You may want to consider the drop impact (if any) on this surface as well.

As the nanoparticles are not bound to the surface, how can you be sure that the water drop does not remove some particles after impact? During bouncing the drop may be contaminated with nanoparticles.

Perhaps some information about the Never Wet product can be provided e.g. an FTIR analysis would be useful or some information from the literature can be given.

Author Response

Rough surfaces of controlled topographies resembling the lotus and rice leaves were fabricated by wire electrical discharge machining (WEDM) on aluminum alloy. Surfaces were then treated with a commercial water repellent product and bouncing dynamics of impact droplets on the fabricated superhydrophobic surfaces was investigated.

This is a very good and meticulous study which can be useful for the readers of the Nanomaterials. I have no major comments on the Technical part which is very good. However, the structure of the manuscript should be improved.

My main concern is that the authors do not compare their results with previously published works. In the Results and Discussion section very few previously published works are cited. However, there are several articles which investigate the impact of bouncing drops on superhydrophobic, lotus-like surfaces e.g. Applied Surface Science 257(21):8857-8863; Science China Physics, Mechanics & Astronomy 2014, Volume 57, Issue 7, pp 1376–1381. These two works are provided as examples. There is a considerable amount of literature on this topic, which is neglected by the authors and it is not included in the Introduction. The authors should include these previously published works in the Introduction describing the novelty of their paper. As the goal of this work is to study the bouncing drops, the novelty cannot be the use of wire electrical discharge machining. More important, the authors should compare (Results and Discussion section) their results with the data of the literature.

Related references have been added, and we have compared their results with the data of the literature in the Results and Discussion section.

Likewise, in the Introduction in the paragraph in which the various techniques are summarised it is better to focus on articles describing the achievement of superhydrophobicity on aluminum surfaces (e.g. Langmuir 2008, vol 24, p 11225 and others) trying to avoid papers which are not related to the substrate of interest. Other works which use nanoparticles for superhydrophobicity can be included.

We have added the related references in the revised manuscript.

The Abstract and particularly the Conclusion sections are not very informative. In the Conclusion section the conclusions should be described more precisely, avoiding general comments such as for instance, “Different surfaces had different effects on the contact time of water droplet.”

The Abstract and Conclusion sections have been rewritten in the revised manuscript.

What is the contact angle of a water drop on Never Wet on polished aluminum? You may want to consider the drop impact (if any) on this surface as well.

The contact angle of a water drop on Never Wet on polished aluminum was 142°, as shown in Figure S1 ("Please see the attachment.") .

As the nanoparticles are not bound to the surface, how can you be sure that the water drop does not remove some particles after impact? During bouncing the drop may be contaminated with nanoparticles.

This is our mistake, we forgot to add the experimental details to the Experimental Section. In fact, prior to testing and characterization of samples, alcohol was used to rinse those nanoparticles that were not bound to the surfaces.

Perhaps some information about the Never Wet product can be provided e.g. an FTIR analysis would be useful or some information from the literature can be given.

Some information about the Never Wet product can be provided, including the EDS and FTIR results.

Round 2

Reviewer 1 Report

The authors revised their manuscript properly, and now it can be accepted.

Author Response

Thank you very much.

Reviewer 2 Report

The questions have been sufficiently addressed. Some additional results have been reported. In this regard, I would point out that the FTIR results quite match the spectrum of SiO2 and the EDS analysis detected the C element; therefore, the coating cannot be composed of silicon alone. I would suggest replacing line 213 as follows: “coating is mainly composed of silicon-based nanoparticles.”.

In Figure 5c, report the peak wavenumbers.

Figs 2a, 2b, 3a, and 3c are already published in other papers. Authors should indicate that each of these images is reproduced (with permission?) from….

Author Response

The questions have been sufficiently addressed. Some additional results have been reported. In this regard, I would point out that the FTIR results quite match the spectrum of SiO2 and the EDS analysis detected the C element; therefore, the coating cannot be composed of silicon alone. I would suggest replacing line 213 as follows: “coating is mainly composed of silicon-based nanoparticles.”.

We have modified in the revised manuscript.

In Figure 5c, report the peak wavenumbers.

We have added peak wavenumbers in Figure 5c.

Figs 2a, 2b, 3a, and 3c are already published in other papers. Authors should indicate that each of these images is reproduced (with permission?) from….

We have added the corresponding contents in the captions of Figures 2 and 3.

Reviewer 3 Report

The revised paper is an improved version which can be published in Nanomaterials.

Author Response

Thank you very much.